# EXPRESSIVENESS IN DEEP REINFORCEMENT LEARNING

## ABSTRACT

Representation learning in reinforcement learning (RL) algorithms focuses on extracting useful features for choosing good actions. Expressive representations are essential for learning well-performed policies. In this paper, we study the relationship between the state representation assigned by the state extractor and the performance of the RL agent. We observe that representations assigned by the better state extractor are more scattered than which assigned by the worse one. Moreover, RL agents achieving high performances always have high rank matrices which are composed by their representations. Based on our observations, we formally define *expressiveness* of the state extractor as the rank of the matrix composed by representations. Therefore, we propose to promote expressiveness so as to improve algorithm performances, and we call it *Expressiveness Promoted DRL*. We apply our method on both policy gradient and value-based algorithms, and experimental results on 55 Atari games show the superiority of our proposed method.

## 1 INTRODUCTION

Deep reinforcement learning (DRL) algorithms, such as DQN (Mnih et al., 2015), A3C (Mnih et al., 2016), DDPG (Lillicrap et al., 2015), and TRPO (Schulman et al., 2015), have been applied in a range of challenging domains including Atari games (Mnih et al., 2015; Schaul et al., 2015; Van Hasselt et al., 2016), robot locomotion tasks (Schulman et al., 2015; 2017) and the game of Go (Silver et al., 2016; 2017). The combination of RL and high-capacity function approximators such as neural networks holds the promise of automating a wide range of decision making and control tasks.

The DRL model often contains two parts. First, a deep neural network is used to extract stateful information from raw signals, e.g, convolutional neural network for images-based games (Mnih et al., 2015; Vinyals et al., 2017) or recurrent neural network for natural language-based games (Narasimhan et al., 2015; Zhao & Eskenazi, 2016). Without any confusions, we call such raw signals as *observation*, call the extracted stateful information as *state* and consider this deep neural network as *state extractor*. Given the representation of state, a feedforward neural network is further used to select different actions to maximize the potential rewards. From the above description, it is easy to see that the performance of an RL agent depends on two parts. First, it depends on whether the state extractor is good. With a good state extractor, the representation which is a depiction of the observation will retain necessary information for taking actions. Second, it depends on the accuracy of the policy: whether the feed-forward model can correctly take the optimal action given the state.

In this paper, we mainly study the relationship between representations extracted by the state extractor and the performance of the RL agents. We observe that when agents achieve better performances, matrices composed by their representations are approximately higher rank.[1] Firstly, we study representations assigned by state extractors in different performed RL agents over all Atari games. We find that representations assigned by the better extractor are more scattered. Furthermore, given different trajectories that lead to high/low rewards as inputs, matrices of representations from the better extractor are always higher rank. Secondly, we find changes of the approximate rank are highly

---

[1]Approximately low rank means that most of the singular values for a matrix are close to zero while only little of them have large values, and higher rank corresponds to more large singular values. In this paper, low/high rank refer to approximately low/high rank.

consistent with changes of rewards, which also demonstrates the positive correlation between the rank and rewards. These observations motivate us to encourage matrices of representations to be high rank during training.

Based on our observations, we formally define *expressiveness* of the state extractor in reinforcement learning. For given markov decision process (MDP) and the initialization state, we can get a realization of the MDP, i.e., a trajectory. The state extractor extracts representations from observations in this trajectory. Representations compose a matrix, which we call it representation matrix. For given MDP with finite state space, expressiveness in RL for the state extractor model is defined as the rank of the representation matrix. According to the definition, expressiveness in RL is both related to the MDP which generates data and the state extractor model. Based on above experimental studies, we can conclude that higher expressiveness will lead to better performances. As we can see that expressiveness is not easy to calculate because the max operator and the infinite number of columns in representation matrix, so we consider the rank of the representation matrix composed by a mini-batch of representations, which is called empirical representation matrix. Empirical expressiveness in RL is defined as the approximate rank of the empirical representation matrix, i.e., if some singular values are approximately zero, we regard them as zero.

We further propose a novel method *ExP* (**Ex**pressiveness **P**romoted) DRL, which aims improve the expressiveness of the state extractor, so as to promote RL algorithm performances. Based on experimental observations and the definition, the empirical representation matrix is encouraged to be high rank in our method. The idea is implemented by adding a regularization term to the loss, which is computationally efficient and can be applied to multiple kinds of DRL algorithms.

We applied our method to A3C (Mnih et al., 2016) and DQN (Mnih et al., 2015). Evaluation results on Atari games show that our method outperforms the baseline on most of games. Furthermore, we also demonstrate that the expressiveness of the state extractor is significantly enhanced by our proposed ExP DRL.

## 2 STATE EXTRACTOR EXPRESSIVENESS

### 2.1 EXPERIMENTAL OBSERVATIONS

In image classification tasks, many papers show that the representation of images (high-level features from top layers) contains useful and abstract information for decision makings, e.g., predicting the categorical labels (Coates et al., 2013). In reinforcement learning, such high-level features will be used to find better actions in pursuit of larger reward.

However, the representation in reinforcement learning is much harder to learn compared to that in supervised learning problems. In image classification task, e.g., ImageNet, different images with the same label contain the same item and most of the images in different categories contain different items. This makes the neural network can learn discriminative information effectively and the learned features are good with intra-class compactness and inter-class separability (Liu et al., 2016). In reinforcement learning, taking shooter games in Atari as an example, the input images which will be fed into the neural network are similar as most of them contains a group of enemies, bullets, the agent and different objects in the game. Based on similar observations, the neural network has to learn to extract fine-grained local features that contain the positions of enemies, the direction of bullets, the current position of the agent and related objects, which are essential for taking actions.

Therefore, to investigate the characteristic of good representations assigned by the state extractor, we compare state extractors in RL models with different performances. We find that representations generated by better state extractor are more scattered compared with worse extractors, and the matrix composed by better representation vectors is higher rank. We observe similar phenomena during training process, which shows that when the matrix formed by representations becomes higher rank, the reward of the RL model become higher. Furthermore, we notice that representations are not always getting more discriminative during training, which also motivates our proposed method.

### 2.1.1 COMPARISON BETWEEN BETTER AND WORSE STATE EXTRACTORS

Comparison between representations assigned by state extractors in RL models with different performances is shown in Fig. 1 and Table 1. We let two models with different model size but same

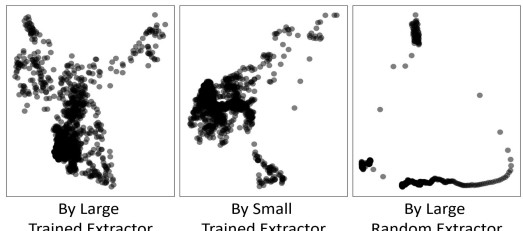

By Large Trained Extractor · By Small Trained Extractor · By Large Random Extractor

| Episode Data | Cumulative Percentage | Average Smallest Ordinal Number of Different Models | | |
|---|---|---|---|---|
| | | Large Trained Model | Small Trained Model | Large Random Model |
| Episode from Large Trained Model | 80% | 164.9 | 137.8 | 103.8 |
| | 90% | 247.7 | 213.8 | 197.4 |
| Episode from Small Trained Model | 80% | 135.4 | 116.3 | 83.7 |
| | 90% | 210.8 | 187.6 | 166.1 |
| Episode from Large Random Model | 80% | 72.1 | 64.7 | 50.9 |
| | 90% | 120.6 | 112.0 | 105.5 |

Figure 1: Two-dimensional embedding of the representations assigned by different state extractors.

Table 1: Relationship between model performance and the smallest ordinal number of singular values.

number of last hidden layer units play the Atari game Gravitar for 200M frames respectively. After training, the larger model achieves 3050 for the game score, while the smaller model gets 600 points. Besides, in order to exclude the bias of the model size, we also observe representations generated by the large model with its randomly initialized weights, which can only gets 0 game point because points are hard to obtained in this game. We refer to these three models with large trained model, small trained model and large random model respectively.

Fig. 1 shows the two-dimensional embedding of the representations in the last hidden layer assigned by three models to game observations in the trajectory played by the large trained model. Plots are generated by using SVD dimension reduction on the matrix composed by representation vectors. Points in the embedding figure of the model with higher final reward are more scattered overall. Observations only have subtle differences on the existence of enemies, the position and direction of the controlled agent. More scattered representations indicate that the better model can distinguish these subtle differences well, and can do better in extracting these fine-grained local features. These scattered representations can benefit policies and thus lead to better performances. In other words, representations assigned by the better state extractor is more discriminative.

To get more accurate and solid observations, we list some statistics over all Atari games in Table 1. For each of the Atari game, representations of a large trained model, a small trained model and a large random model are compared. To study the relation between performances and representations, we select games in which the large trained model performs best and the large random model performs worst. In this way, 39 games among 55 games are picked. Then for each game, we generate three trajectories with different final reward by interacting with the environment using three models. Observations in these three trajectories are sent to three state extractor models respectively, and then 9 matrices composed of generated representations are obtained for every game. We sort singular values of the matrix from largest to smallest, and calculate the smallest ordinal number while the cumulative percentage of singular values is above a certain threshold. We list average smallest ordinal numbers of selected 39 games for threshold 80% and 90% in Table 1. These data demonstrate that given different trajectories and model size, the average smallest ordinal number always positively correlates with the model performance: the smallest ordinal number of the better model is always larger then which of the the worse model. This means that the matrix composed of representations generated by the better model is always higher rank than which generated by the worse model, which suggests that regardless of input trajectories and model size, states assigned by the better extractor is more discriminative and expressive comparing with the worse model, because higher rank corresponds to better discrimination. Representations which are not expressive enough cause troubles for decision making. Therefore, representations should be encouraged to be more discriminative and expressive for obtaining good policies.

### 2.1.2 TRENDS DURING TRAINING

In order to investigate the found phenomena extensively, and also make the observation more convincing, we study representation changes during training in this section. We train a model to play the Atari game WizardOfWor for 200M frames. Each time the model is updated with a mini-batch of transitions, representation vectors in the last hidden layer are regarded as a matrix, and singular values of this matrix are recorded. We calculate the smallest ordinal number while the cumulative percentage of singular values is above a certain threshold (i.e., 80%) as last section. Large ordinal

number means that the matrix composed by representation vectors is high rank, and corresponding representations are discriminative.

Two curves are plotted in Fig.2. The blue one tracks testing rewards, and the yellow one tracks smallest ordinal numbers. It shows that curve trends of testing rewards and smallest ordinal numbers are highly consistent: when smallest ordinal numbers increase/decrease, rewards also become high/low. Same phenomena on more games can also been seen in Fig. 6. This consistency demonstrates that states assigned by the better extractor, which form high rank matrices, are more discriminative and expressive, and holding discriminative representations is necessary for learning good policies. Besides these same conclusions mentioned in last section, we also notice that during training, the smallest ordinal number is not always increase. But representations which are not expressive enough cause troubles for decision making. Thus, the expressiveness of state extractors should be promoted (i.e., formed matrices should be encourage to be high rank) during training.

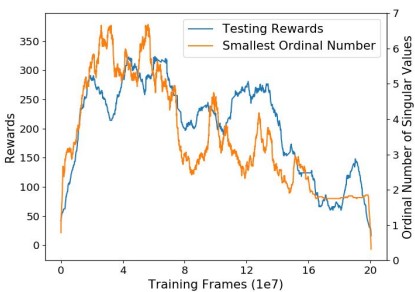

Figure 2: Curves tracking testing rewards and smallest number of singular values during training process. Curves are smoothed.

## 2.2 DEFINITION OF THE EXPRESSIVENESS

Based on previous observations, we formally define a new concept *expressiveness* in RL in this section. Consider an Markov decision process (MDP) $\mathcal{M} = (\mathcal{S}, \mathcal{A}, p, \gamma, r)$, where $\mathcal{S}$ denotes the observation space, $\mathcal{A}$ denotes the action space, $p : \mathcal{S} \times \mathcal{A} \rightarrow \mu(\mathcal{S})$ denotes the transition probability with $\mu(\mathcal{S})$ denoting the space of measures on $\mathcal{S}$, $\gamma \in (0, 1)$ denotes the discount factor and $r : \mathcal{S} \times \mathcal{A} \rightarrow \mathbb{R}$ is the reward function. The goal of reinforcement learning is to learn a policy $\pi(x)$ that maximize the average future reward function $\mathbb{E}_{\pi(x), p(x,a)}[\sum_{k=0}^{\infty} \gamma^k r(x_k, a_k)|x_0]$.

Assume that $\{x_1, \cdots, x_b\}$ is a mini-batch of observations. We use operator $h(x) : \mathcal{X} \rightarrow \mathbb{R}^d$ to denote the state extractor. For given observation $x_i$, we obtain its the representation $h(x_i) = (h_{i1}, \cdots, h_{id})$. The representations of a mini-batch of observation consist a matrix, which we call it representation matrix and denote it as $H = (h(x_1), \cdots, h(x_b))^T = \{h_{ij}\}_{i=1,\cdots,b;j=1,\cdots,d}$, where $h_{ij}$ denotes the $j$-th representation calculated using the $i$-th observation in the mini-batch.

**Definition 1.** *(Expressiveness in RL) For given MDP $\mathcal{M} = (\mathcal{S}, \mathcal{A}, p, \gamma, r)$ with finite observation space $\mathcal{S}$ and initial observation $x_0$, the expressiveness $\mathcal{E}_{\mathcal{M}}(h)$ for extractor model $h$ is defined as the rank of matrix composed by $h(X_t)$, $\forall t$, i.e.,*

$$\mathcal{E}_{\mathcal{M}}(h) = rank\{h(X_1), \cdots, h(X_t), \cdots\}. \tag{1}$$

The defined expressiveness is related to both the MDP $\mathcal{M}$ and the model $h$. For fixed MDP, the expressiveness is similar to that is defined in supervised learning which is related to the function approximation ability. For fixed $h$, the expressiveness is related to the MDP $\mathcal{M}$. Consider two two MDP $\mathcal{M}_1$ and $\mathcal{M}_2$ with $|\mathcal{S}_1| = |\mathcal{S}_2|$. If $\mathcal{M}_1$ is ergodic and $\mathcal{M}_2$ is non-ergodic (i.e., it can only visit a subset of observations), then the matrix $\{h(X_1), \cdots, h(X_t), \cdots\}$ will more possibly be low rank.

In practical, it is not easy to exactly calculate the rank for a trajectory with infinite length. Thus, we define the following empirical expressiveness.

**Definition 2.** *(Empirical expressiveness in RL) For given observations $\{x_1, \cdots, x_b\}$ and representation matrix $H_{b \times d}$, we order singular values of $H$ from large to small as $\sigma_1(H), \cdots, \sigma_{\min\{b,d\}}(H)$. The empirical expressiveness $\mathcal{E}_{b,\epsilon}(h)$ for extractor model $h$ is defined as*

$$\mathcal{E}_{b,\epsilon}(h) = argmin_j \left\{ j : \frac{\sum_{i=1}^{j} \sigma_i(H)}{\sum_{i=1}^{\min\{b,d\}} \sigma_i(H)} > 1 - \epsilon \right\}. \tag{2}$$

Please note that the empirical expressiveness is related to the number of sampled observations, given precision $\epsilon$ and the extractor model $h$. Please note that the rank of the representation matrix is full in practical because the singular values will not be exactly equal to zero. So we introduce $\epsilon$ in the definition.

## 3 ExP (Expressiveness Promoted) DRL

We propose our method named ExP DRL, which intends to improve the expressiveness of the state extractor, so as to promote DRL algorithm performances. Based on the new concept expressiveness, experimental observations in Sec. 2.1 can be summarized as: state extractors with better expressiveness lead to better policies, and the expressiveness not always becomes better during training. Hence, in order to obtain better policies, we propose to promote the expressiveness of the state extractor, which means encouraging matrices formed by extracted representations to be high rank.

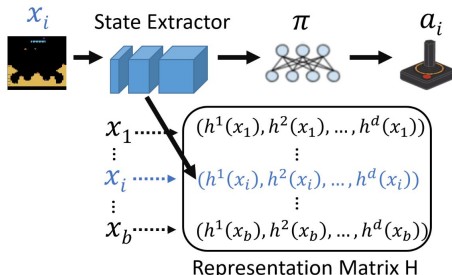

Figure 3: General architecture of a DRL model and its representation matrix $H$. $u$ is the number of hidden units, and $b$ is the mini-batch size.

To be specific, the general architecture of a DRL model and its representation matrix are shown in Fig. 3. The state extractor extract features from the observation $x$ to generate the representation vector $h(x)$, then these representations are sent to a feedforward neural network to select the action $a$. For one observation $x_i$, a representation vector $h(x_i)$ is generated, and for a mini-batch of observations, $\{h(x_1), ..., h(x_b)\}$ of size $b$, these vectors compose a representation matrix $H$. In ExP DRL, this matrix $H$ is encouraged to be high-rank.

To minimize computation costs, we adopt a simple way to encourage high rank representation matrices. A regularization term is added to the policy loss, making matrix $H$ to be high rank. Thus, total loss is denoted as,

$$\mathcal{L} = \mathcal{L}_{policy} + \alpha * R(H), \tag{3}$$

where $\alpha$ is the coefficient. According to description in Section 2.2, $R(H)$ should be equal to $-\mathcal{E}_{b,\epsilon}(h)$ in order to encourage $H$ to be high rank. Directly optimize $-\mathcal{E}_{b,\epsilon}(h)$ is not easy because of the argmax operator, thus we propose three types of regularizers which can reflect the scale of $-\mathcal{E}_{b,\epsilon}(h)$ to some extent.

**Negative Nuclear Norm** Minimizing the nuclear norm is usually used as a constrain in low-rank matrix completion problems (Cai et al., 2010). The nuclear norm of a matrix is the sum of all singular values of this matrix, which is defined as

$$||H||_1 = \sum_{i=1}^{\min\{b,d\}} \sigma_i(H), \tag{4}$$

where $\sigma_i(H)$ is the $i$th singular value of $H$. Here we try to make the representation matrix to be high rank. Hence, we add a negative nuclear norm to the loss, so $R(H) = -||H||_1$.

**Max Minus Min** Besides enlarging all singular values, a more direct way to improve expressiveness of the state extractor is to reduce the gap between the maximum and the minimum singular value. This gap is defined as,

$$\mathcal{G}(H) = \sigma_{max}(H) - \sigma_{min}(H), \tag{5}$$

and $R(H) = \mathcal{G}(H)$.

**Condition Number** The aforementioned two kinds of rank regularization terms, $||H||_1$ and $\mathcal{G}(H)$, are sensitive with the scale of singular values. So in order to reduce this bias, we also try to use the condition number to be the regularization term.

Condition number is defined as the ratio of the maximum and the minimum singular value:

$$\mathcal{K}(H) = \frac{\sigma_{max}(H)}{\sigma_{min}(H)}. \tag{6}$$

The scale bias can be resolved by the division operator in the condition number. Similar with the *Max Minus Min* term, the condition number should be minimized during training to improve the expressiveness, so $R(H) = \mathcal{K}(H)$ here.

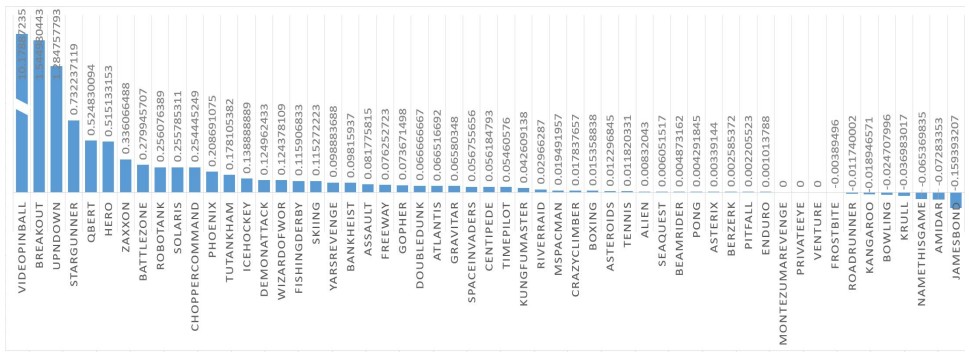

Figure 4: Improvements of our method ExP DRL compared to A3C, using the metric given in Eq. 7.

In summary, the key improvement in ExP DRL is that we change the loss function to Eq. 3. And we introduce three kinds of regularization terms: the negative nuclear norm $-||H||_1$, the max minus min $\mathcal{G}(H)$ and the condition number $\mathcal{K}(H)$. We analyze differences among these 3 terms in experiments.

Note that ExP DRL can be applied to various kinds of DRL algorithms. The expressiveness can be promoted by simply adding a regularization term when the RL model is updated. Excepting these transitions used for updating the RL model, no other data are introduced for encouraging highly expressive representations. Thus, our method can be widely applied to current DRL algorithms, with little extra computational cost.

## 4 EXPERIMENTS

The experiment section is designed to answer these questions: (1) Can algorithm performances be improved by applying our proposed expressiveness promoted method? (2) What is the difference between 3 proposed rank regularization terms? (3) Is the expressiveness of the state extractor promoted after using ExP DRL? (4) Can ExP DRL improve performances of multiple kinds of DRL alogirithms? These questions are answered respectively in 4 subsetions below.

### 4.1 OVERALL PERFORMANCES ON A3C

#### 4.1.1 SETTINGS

We evaluate our proposed method on 55 Atari games (Bellemare et al., 2013). Atari game learning environment is one of the most popular and challenging RL task because of its high-dimensional and diverse observations. Here we use OpenAI Gym (Brockman et al., 2016) package.

In this section, we use A3C (Mnih et al., 2016) as our baseline, and use the pytorch-a3c package (Kostrikov, 2018), which is also be adopted by Peysakhovich & Lerer (2017), to implement. The network architecture is shown in Appendix A. An environment wrapper is utilized to simplify original visual screens, and they are processed to $42 \times 42$ gray-scale images. We use a frame-skip of 4 here. The number of processes is set to be 16. Besides, each agent is trained for 200M game frames, and the obtained reward is averaged over 5 runs with different random seeds.

For our proposed method, outputs of the LSTM are taken as representations generated by the state extractor. Each time the A3C model is updated using a mini-batch of transitions, representations of observations in these transitions form the matrix $R$. To improve expressiveness, this matrix is encouraged to be high rank by adding the regularization term. Expect the coefficient $\alpha$, other hyper parameters and settings are set as same as the baseline.

#### 4.1.2 PERFORMANCES

Following the previous work (Wang et al., 2015), we use the measure below to compare the performance of ExP DRL over the baseline.

$$\frac{\text{Score}_{\text{Agent}} - \text{Score}_{\text{Baseline}}}{\max\{\text{Score}_{\text{Human}}, \text{Score}_{\text{Baseline}}\} - \text{Score}_{\text{Random}}} \times 100\%. \tag{7}$$

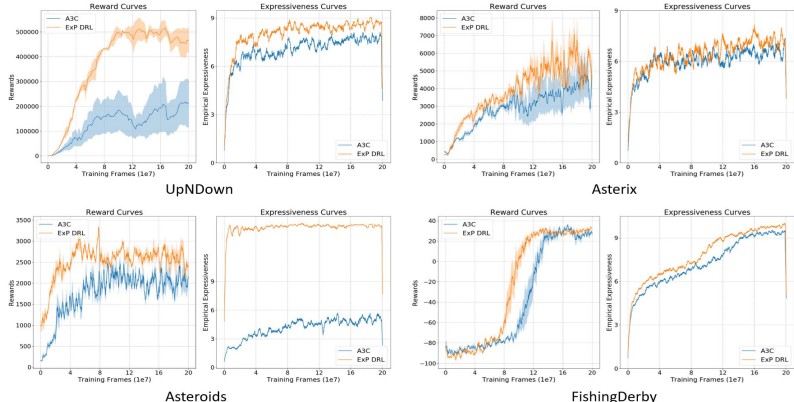

Figure 6: Testing rewards curves (left) and empirical expressiveness curves (right) on 4 Atari games for ExP DRL (yellow) and the baseline (blue).

We summarize the improvement of our method over the A3C baseline in Fig. 4, and all raw scores and normalized socres are listed in Table 5 in the appendix. Among 55 games, ExP DRL outperforms the baseline for 43 games. These demonstrate that encouraging representation matrices to be high rank promotes performances of original algorithms. Improving expressiveness of state extractors benefits learning good policies.

## 4.2 THREE RANK REGULARIZATION TERMS

To compare different regularization terms, we test each term on the same game with same coefficient $\alpha$. We empirically set $\alpha$ as 0.01 here. The way to choose $\alpha$ and an ablation study of $\alpha$ are shown in Appendix B. Performances of UpNDown and Qbert are plotted in Fig. 5. In general, the algorithm performance can be promoted largely by applying ExP DRL. And improvements made by Max Minus Min and Condition Number is larger than which made by Negative Nuclear Norm. This is reasonable because that enlarging all singular values may not increase the expressiveness of the state extractor. It is not certain that which singular value gets larger when applying the Negative Nuclear Norm. For example, the expressiveness will decrease when only the largest singular value becomes larger. Instead, the object of Max Minus Min and Condition Number is to reduce the gap between the largest and the smallest singular value, which directly enlarges the expressiveness of state extractors.

We use Max Minus Min and Condition Number with $\alpha$ as 0.01 to cover results of most games in the previous section. All hyper parameters we used are listed in Table 4 in the appendix.

## 4.3 EXPRESSIVENESS ANALYSIS

In this section we analysis the expressiveness of the state extractor learned using the baseline and our proposed method. Curves tracking testing rewards and the empirical expressiveness are plotted in Fig. 6. From testing reward curves we can see that ExP DRL outperforms the A3C baseline on these games. These results show that improving expressiveness significantly benefits policy learning.

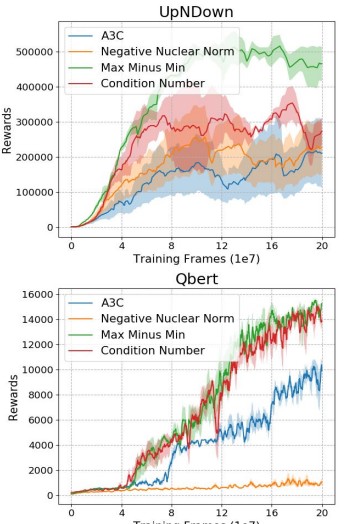

Figure 5: Rewards of ExP DRL applying three regularization terms. Shaded areas depict variances.

Combining testing reward figures and expressiveness figures, we can observe that: (1) the empirical expressiveness of the state extractor is increased by adding the regularization term, which demonstrate that the proposed method can really enhance the expressiveness. (2) Curve trends of the reward and the expressiveness are highly consistent. This is same as what

we observe in Sec. 2.1.2. High rewards and high expressiveness state extractors come out together, which imply that holding good representations is necessary for learning good policies.

## 4.4 PERFORMANCES ON DQN

Since our method ExP DRL does not contain any algorithm related operations, we investigate whether it can promote performances of another kind of DRL algorithm. Experiments on policy gradient algorithms have been done based on A3C in previous sections, so we choose Q-learning here, and use DQN (Mnih et al., 2015) as our baseline.

| Metrics | Average Last Reward |
|---|---|
| #games perform better | 20 |
| #games perform worse | 9 |
| #games perform same | 1 |

Table 2: Performances of 30 Atari games. Average last reward is the average model testing reward after training over 5 runs.

We use same network architecture and hyper-parameters as Mnih et al. (2015). We run 30 Atari games (first 30 in alphabetical order) and run each game for 200M frames. All socres are listed in Table 6 in the appendix, and we summarize the overall performances of these 30 Atari games in Tab. 2. Results show that ExP DRL outperforms the baseline in most of games. This demonstrates the superiority of ExP DRL over the DQN baseline, and supports our claim that encouraging highly expressive state extractor can promote performances of multiple kinds of DRL algorithms.

## 5 RELATED WORKS

State representation learning in RL has been studied in many research works. It focus on learning features which can capture useful information for taking good actions (Lesort et al., 2018). In some of proposed methods, auxiliary models, including variational auto-encoder (van Hoof et al., 2016), denoising auto-encoder (Higgins et al., 2017), auto-encoder (Mattner et al., 2012), Generative Adversarial Networks (Donahue et al., 2016; Shelhamer et al., 2016) and some other models (Oh et al., 2017; Weber et al., 2017), are built for refining the representation learning. These models may be designed as a part of the network architecture, and are trained end-to-end (Oh et al., 2017; Pathak et al., 2017; Tamar et al., 2016), or they can also be trained separately and are used for helping decision making (Weber et al., 2017). These auxiliary models help to improve representation learning via completing some certain tasks, such as reconstructing current observation or state (Watter et al., 2015), predicting future observations or states (Oord et al., 2018; François-Lavet et al., 2018; Munk et al., 2016), and recovering actions given transitions (Zhang et al., 2018). Some essential information for taking actions can be retained by completing these auxiliary tasks. Different with these methods, none other models need to be built and trained in our method. We improve the expressiveness of the state extractor by simply adding a regularization term.

Task specific prior knowledge or information are utilized to improve representation learning in some proposed methods. For example, detection of moving objects is used in Goel et al. (2018) for better learning video games. Jonschkowski & Brock (2015) proposes to use robotic prior knowledge for robot learning. In some chatting systems (Zhao & Eskenazi, 2016), task-related information, such as extracted named entities, are utilized for dialog state representation learning. For our method, we focus on the expressiveness of the state extractor, which does not contain task-specific informations.

Besides, there are some works discussing expressiveness in deep learning (Raghu et al., 2016; Cohen et al., 2016). These works focus on the power of the model, but not internal representations. In addition, out basic setting is for DRL algorithms, which is different with these works.

## 6 CONCLUSIONS

In this paper, we mainly study the relationship between representations extracted by the state extractor and the performance of the RL agents. We observe that when RL agents achieving high rewards, its representations become discriminative, and the representation matrix goes to be high rank. Therefore, we formally define the expressiveness of the state extractor as the rank of the representation matrix. We then further propose a new method ExP DRL, in which algorithm performances are promoted via improving the expressiveness. Experiments of A3C on 55 Atari games and DQN on 30 games demonstrate that ExP DRL can promote their performances significantly.

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

## A  MODEL ARCHITECTURE

The model architecture used for A3C experiments is shown in Fig. 7. We use a relatively small model here comparing with the original A3C paper. It is hard to train with the full architecture described in the paper for us because of the computation resource limits.

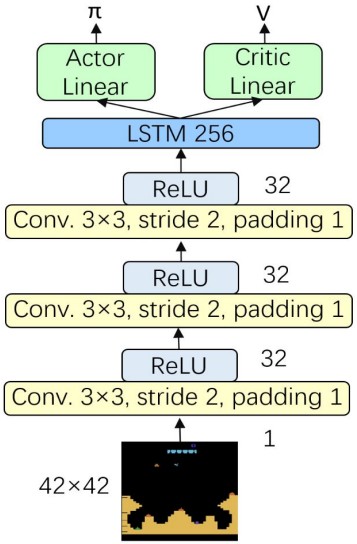

Figure 7: Model architecture.

## B  HYPER PARAMETERS AND ABLATION STUDY ON $\alpha$

The hyper parameter $\alpha$ is chosen as following: 1) we first choose the best alpha for two games from 1 to $10^{-6}$; 2) then we apply the best alpha to all games and found that it performs well in most of the games; 3) next we fine tune the games which are not perform well. All hyper parameters used in A3C experiments are listed in Table 4.

We conduct an ablation study on $\alpha$ to investigate the effectiveness of the proposed ExP DRL. We select 10 games and fix their rank regularization term type. Then we run ExP DRL with different $\alpha$ (from $10^{-1}$ to $10^{-6}$). Performances are evaluated using the metric in Eq. 7, and results are listed in Table 3. These show that our proposed ExP DRL outperforms the baseline on most of games with most of $\alpha$, which demonstrates that ExP DRL is effective. The rank regularization term plays a major role on performance improvements, and $\alpha$ is used for tuning.

| Game Name | $\alpha$ | | | | | |
|---|---|---|---|---|---|---|
| | 0.1 | 0.01 | 0.001 | 0.0001 | 0.00001 | 0.000001 |
| VideoPinball | 10.17887 | 0.945665 | 10.94677 | 0 | 0 | 0 |
| Breakout | 2.027379 | 1.54498 | 2.394444 | 0.556113 | 1.964196 | 2.358339 |
| UpNDown | 0.143224 | 1.284758 | 1.170465 | 1.625934 | -0.93254 | 0.380775 |
| StarGunner | -0.36373 | 0.732237 | 0.579887 | 0.173902 | 0.055844 | -0.26138 |
| Qbert | 0.433576 | 0.52483 | 0.522963 | 0.66813 | 0.417472 | 0.184318 |
| Hero | 0.318271 | 0.322714 | -0.27708 | 0.515133 | 0.002605 | 0.334302 |
| Zaxxon | -0.36779 | 0.336066 | 0.056127 | 0.72826 | 0.328048 | -0.33572 |
| BattleZone | 0.868114 | 0.279946 | 0.230932 | 0.181918 | 0.291257 | 0.189458 |
| Robotank | 0.407986 | 0.256076 | 0.416667 | 0.351562 | 0.138889 | 0.368924 |
| Solaris | 0.255785 | 0.284659 | -0.00198 | 0.002968 | -0.00099 | -0.00099 |
| Average | 1.390169 | 0.651193 | 1.603919 | 0.480392 | 0.226478 | 0.321802 |

Table 3: Ablation Study on $\alpha$.

| Game Name | Rank Regularization Term | $\alpha$ |
|---|---|---|
| Alien | Condition Number | 0.01 |
| Asteroids | Condition Number | 0.01 |
| Berzerk | Condition Number | 0.01 |
| Breakout | Condition Number | 0.01 |
| CrazyClimber | Condition Number | 0.01 |
| DoubleDunk | Condition Number | 0.01 |
| IceHockey | Condition Number | 0.01 |
| Krull | Condition Number | 0.01 |
| MsPacman | Condition Number | 0.01 |
| NameThisGame | Condition Number | 0.01 |
| Phoenix | Condition Number | 0.01 |
| Pong | Condition Number | 0.01 |
| RoadRunner | Condition Number | 0.01 |
| Robotank | Condition Number | 0.01 |
| TimePilot | Condition Number | 0.01 |
| WizardOfWor | Condition Number | 0.01 |
| Zaxxon | Condition Number | 0.01 |
| Frostbite | Condition Number | 0.0001 |
| Gopher | Condition Number | 0.0001 |
| Hero | Condition Number | 0.0001 |
| Bowling | Condition Number | 0.00001 |
| KungFuMaster | Condition Number | 0.00001 |
| Assault | Max Minus Min | 0.01 |
| Asterix | Max Minus Min | 0.01 |
| Atlantis | Max Minus Min | 0.01 |
| BankHeist | Max Minus Min | 0.01 |
| BattleZone | Max Minus Min | 0.01 |
| BeamRider | Max Minus Min | 0.01 |
| Boxing | Max Minus Min | 0.01 |
| ChopperCommand | Max Minus Min | 0.01 |
| DemonAttack | Max Minus Min | 0.01 |
| Enduro | Max Minus Min | 0.01 |
| FishingDerby | Max Minus Min | 0.01 |
| MontezumaRevenge | Max Minus Min | 0.01 |
| Pitfall | Max Minus Min | 0.01 |
| Qbert | Max Minus Min | 0.01 |
| Riverraid | Max Minus Min | 0.01 |
| Skiing | Max Minus Min | 0.01 |
| SpaceInvaders | Max Minus Min | 0.01 |
| StarGunner | Max Minus Min | 0.01 |
| Tennis | Max Minus Min | 0.01 |
| UpNDown | Max Minus Min | 0.01 |
| Venture | Max Minus Min | 0.01 |
| YarsRevenge | Max Minus Min | 0.01 |
| Gravitar | Negative Nuclear Norm | 1 |
| Centipede | Negative Nuclear Norm | 0.1 |
| Freeway | Negative Nuclear Norm | 0.1 |
| Solaris | Negative Nuclear Norm | 0.1 |
| VideoPinball | Negative Nuclear Norm | 0.1 |
| Amidar | Negative Nuclear Norm | 0.001 |
| Kangaroo | Negative Nuclear Norm | 0.0001 |
| PrivateEye | Negative Nuclear Norm | 0.0001 |
| Seaquest | Negative Nuclear Norm | 0.0001 |
| Tutankham | Negative Nuclear Norm | 0.0001 |
| Jamesbond | Negative Nuclear Norm | 0.00001 |

Table 4: Hyper Parameters Used in A3C Experiments.

## C  GAME SCORES

Raw scores and normalized scores in A3C and DQN experiments are listed in Table 5 and Table 6 respectively. And the normalized score is calculated as follow (Wang et al. (2015)):

$$\frac{\text{Score}_{\text{Agent}} - \text{Score}_{\text{Baseline}}}{\text{Score}_{\text{Human}} - \text{Score}_{\text{Random}}} \times 100\%. \tag{8}$$

Human scores and random scores are taken from Wang et al. (2015). We only list scores of 27 games here, because the human score and random score of other 3 games are unavailable.

| Game Name | Raw | | Normalized | |
|---|---|---|---|---|
| | baeline | rank | baeline | rank |
| Alien | 999.4444444 | 1051.388889 | 14% | 15% |
| Amidar | 539.3055556 | 427.9722222 | 35% | 27% |
| Assault | 735.5277778 | 782.0277778 | 123% | 133% |
| Asterix | 4491.666667 | 4516.666667 | 59% | 59% |
| Asteroids | 2013.888889 | 2452.222222 | 3% | 4% |
| Atlantis | 3121788.889 | 3328544.444 | 23706% | 25283% |
| BankHeist | 1000.833333 | 1096.944444 | 157% | 173% |
| BattleZone | 16861.11111 | 25111.11111 | 45% | 73% |
| BeamRider | 3724.555556 | 3796.222222 | 24% | 24% |
| Berzerk | 592.7777778 | 598.0555556 | 19% | 20% |
| Bowling | 44.58333333 | 41.83333333 | 8% | 6% |
| Boxing | 97.97222222 | 99.5 | 896% | 910% |
| Breakout | 112.3888889 | 283.5555556 | 421% | 1072% |
| Centipede | 4709.083333 | 5180.833333 | 33% | 39% |
| ChopperCommand | 7202.777778 | 9311.111111 | 79% | 105% |
| CrazyClimber | 112894.4444 | 114741.6667 | 444% | 452% |
| DemonAttack | 4478.472222 | 5012.083333 | 132% | 149% |
| DoubleDunk | -1 | 0 | 938% | 1000% |
| Enduro | 0 | 0.833333333 | 10% | 10% |
| FishingDerby | 23.55555556 | 35.22222222 | 122% | 137% |
| Freeway | 6.25 | 8.194444444 | 24% | 32% |
| Frostbite | 286.1111111 | 270 | 5% | 5% |
| Gopher | 5770 | 6176.666667 | 268% | 288% |
| Gravitar | 37.5 | 226.3888889 | -7% | -1% |
| Hero | 20781.80556 | 33278.47222 | 79% | 131% |
| IceHockey | -2.944444444 | -1.527777778 | 66% | 80% |
| Jamesbond | 538.8888889 | 458.3333333 | 151% | 127% |
| Kangaroo | 2375 | 2325 | 86% | 84% |
| Krull | 8533.666667 | 8260.666667 | 771% | 743% |
| KungFuMaster | 36094.44444 | 37619.44444 | 175% | 182% |
| MontezumaRevenge | 0 | 0 | -1% | -1% |
| MsPacman | 1282.777778 | 1578.611111 | 7% | 9% |
| NameThisGame | 6795 | 6465 | 100% | 93% |
| Phoenix | 5339.722222 | 6498.333333 | 76% | 97% |
| Pitfall | -14 | 0 | 5% | 5% |
| Pong | 20.83333333 | 21 | 116% | 116% |
| PrivateEye | 100 | 100 | -1% | -1% |
| Qbert | 7697.916667 | 13944.44444 | 63% | 116% |
| Riverraid | 12343.88889 | 12753.05556 | 85% | 88% |
| RoadRunner | 58405.55556 | 57722.22222 | 872% | 861% |
| Robotank | 28 | 34.55555556 | 394% | 495% |
| Seaquest | 1789.444444 | 2032.777778 | 4% | 5% |
| Skiing | -9237.166667 | -7899.916667 | 52% | 64% |
| Solaris | 17.77777778 | 2316.111111 | -23% | 3% |
| SpaceInvaders | 589.1666667 | 661.9444444 | 32% | 37% |
| StarGunner | 26861.11111 | 46019.44444 | 296% | 513% |
| Tennis | -0.25 | 0 | 144% | 146% |
| TimePilot | 11513.88889 | 11963.88889 | 347% | 366% |
| Tutankham | 170.2222222 | 198.2777778 | 125% | 148% |
| UpNDown | 204897.2222 | 467231.9444 | 2222% | 5077% |
| Venture | 0 | 0 | -2% | -2% |
| VideoPinball | 0 | 14362.38889 | -1152% | -134% |
| WizardOfWor | 66.66666667 | 533.3333333 | -20% | -7% |
| YarsRevenge | 16634.27778 | 21149.13889 | 33% | 43% |
| Zaxxon | 8375 | 11052.77778 | 99% | 133% |
| **Mean** | | | 596% | 721% |
| **Median** | | | 76% | 88% |

Table 5: Raw scores and normalized scores for all games in A3C experiments.

| | Raw | | Normalized | |
|---|---|---|---|---|
| Game Name | baeline | rank | baeline | rank |
| Alien | 274.0164 | 722.1259 | 2% | 10% |
| Amidar | 907.5578 | 794.7486 | 59% | 51% |
| Assault | 2555.212 | 2673.75 | 517% | 543% |
| Asterix | 1272.168 | 1002.487 | 15% | 11% |
| Asteroids | 261.4554 | 1061.01 | -2% | 1% |
| Atlantis | 1337938 | 2090013 | 10101% | 15837% |
| BankHeist | 562.3635 | 615.295 | 87% | 95% |
| BattleZone | 24921.86 | 4500 | 72% | 3% |
| BeamRider | 14826.56 | 15418.49 | 99% | 103% |
| Berzerk | 444.1128 | 632.8937 | 12% | 21% |
| Bowling | 29.43 | 30 | -5% | -5% |
| Boxing | -49.3083 | -8.77755 | -431% | -66% |
| Breakout | 262.8333 | 249.875 | 993% | 944% |
| Centipede | 1640.954 | 5106.617 | -3% | 38% |
| ChopperCommand | 1522.015 | 4147.24 | 11% | 42% |
| CrazyClimber | 450.7576 | 483.6859 | -38% | -38% |
| DemonAttack | 2288.795 | 6179.065 | 64% | 185% |
| DoubleDunk | -1.25 | -0.75 | 922% | 953% |
| Enduro | 10.01563 | 7.32 | 11% | 11% |
| FishingDerby | -10.6449 | 15.56085 | 81% | 113% |
| Freeway | 32.76923 | 32.78659 | 128% | 128% |
| Frostbite | 133.3333 | 160 | 2% | 2% |
| Gopher | 2751.741 | 12247.29 | 121% | 582% |
| Gravitar | 332.663 | 263.7448 | 3% | 1% |
| Hero | 302.9514 | 37 | -5% | -6% |
| IceHockey | -8.36886 | -11.6054 | 13% | -19% |
| Jamesbond | 888.6883 | 790.4797 | 255% | 226% |
| **Mean** | | | 485% | 732% |
| **Median** | | | 15% | 38% |

Table 6: Raw scores and normalized scores for all games in DQN experiments.

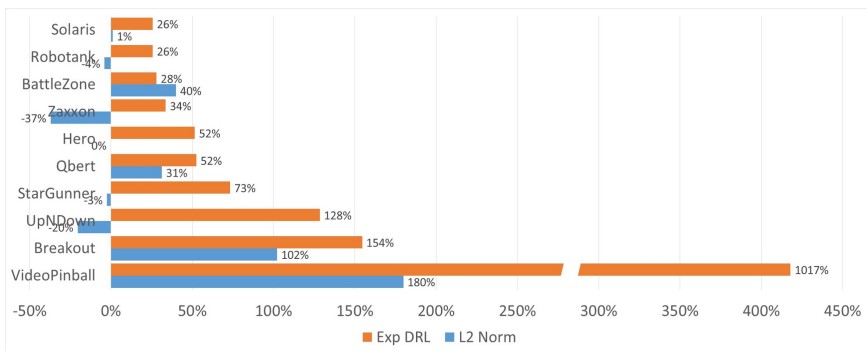

Figure 8: Improvement comparison over baseline A3C between L2 norm regularizer and our method ExP DRL.

## D  USING L2 NORM REGULARIZER

In this section, we run A3C with L2 norm regularizer on 10 Atari games, and performances are compared with performances of our proposed ExP DRL. We tune the coefficient of L2 norm regularizer (denoted as $\beta$ here) in a similar way with what we used for $\alpha$. We firstly choose best three $\beta$ for 2 games from 1 to $10^{-8}$. These best three $\beta$ are $10^{-4}, 10^{-5}$ and $10^{-6}$. Then we apply these three $\beta$ to rest of 8 games, and choose the best performed one to repeat 4 times additionally. Thus the final reward is averaged over 5 times. We keep other settings same as Sec. 4.1.1.

Using the measure in Eq. 7, we list performances of our proposed ExP DRL and L2 norm in Fig. 8. Among 10 games, ExP DRL outperforms L2 norm regularizer in 9 games. These results demonstrate that our method ExP DRL is more efficient than L2 norm regularizer. Besides, adding L2 norm regularizer not always improves performances. It seems that it has an uncertain effect on final performances. Our method and the L2 norm regularizer influence the final performance in a different way. The L2 norm prevents overfitting, and ExP DRL improves expressiveness of the state extractor, leading to better performed RL models.

