# OpenReview forum: "Expressiveness in Deep Reinforcement Learning"
_ICLR.cc/2019/Conference_

### Official Review · AnonReviewer2 · 2018-11-01
**Interesting results, but "insights" seem misleading**

**Rating:** 4
**Confidence:** 4

**Review:**

The authors propose the notion of "expressiveness" of state representation by simply checking the effective rank of the state vectors formed by a sampled trajectory. The authors then propose a regularizer that promotes this expressiveness. Experiments on a large set of Atari games show that this regularizer can improve the performance of reinforcement learning algorithms.

1. The authors motivate this notion through the example in section 2 where the model with higher capacity performs better. The authors note that the learned state matrix of the higher-capacity model has higher rank. But this is completely expected and well-known in machine learning in the regime where the models have insufficient capacity. Imagine the case where the true state representation consists of only a two-dimensional vector (say, the location vector), and an image is produced through a linear map. Now suppose we try to learn a K-dimensional state representation for K>>2 and promote a high-rank matrix. What do we expect to get from this?

2. It seems that promoting a small gap between the singular values gives a performance improvement in the case of Atari games. One can easily interpret this as a regularizer that simply prevents overfitting by equalizing the magnitudes in most parameter values. In this sense, I do not see a fundamental difference between this regularizer and say, the L2 norm regularizer. Did the authors try comparing this with an L2 norm regularizer?


The authors present very interesting empirical results, but I am not convinced that the proposed notion of "expressiveness" properly explains the performance improvement in this set of tasks. I am also not convinced that this is the right notion to promote in general. In this sense, I am afraid I cannot recommend acceptance.

---

> ### Author Response · Authors · 2018-11-25
> **To Reviewer3**
>
> Thank you for your helpful comments. The following is our responses.
>
> 1. We have added an experiment in the section 2.1 to show that higher-capacity model is not necessarily to have higher rank. We compare the ranks of the representations generated by three networks: NN(1) and NN(2) with the same network size and NN(3) with smaller network size. The performance ranking for the three models is: NN(1)>NN(3)>NN(2). We observed that the representation’s ranks of them follow the same rank: rank(H_{NN(1)})>rank(H_{NN(2)})>rank(H_{NN(3)}). It indicates that representation’s rank is highly related to the performance of RL and less related to the network size or model capacity.
>
> 2. Promoting high-rank representation will not bring negative influence for learning the linear classifier because “Linearly independent features are linearly separable [1]”. For deep neural networks, it is always hard to explain what the exact meaning of the representation vector. As we observed that the empirical expressiveness of the representation matrix is worse in Atari games and in the training process it increases, we choose to promote the rank to help Atari games to quickly converge and achieve better performance.
>
> 3. We have added the comparison with L2 norm regularizer in Appendix D. The results show that adding L2 norm regularizer helps less. Please note that reinforcement tasks are very different from deep learning tasks. Generalization ability is well defined and very important for DL tasks, but for RL, it is unclear which measure is the directly related to the performance. So we propose one measure “expressiveness” and verify its reasonability.
>
> [1]“Optimization Landscape and Expressivity of Deep CNNs” https://arxiv.org/pdf/1710.10928.pdf

---

> > ### Comment · AnonReviewer2 · 2018-12-07
> > **Final comments**
> >
> > I appreciate the authors' effort in revising the paper and including additional experimental results. However, my main concern (#1) remains. The revised section 2 did not address the key question here: if a simple low-rank state representation is sufficient for the problem (e.g. capable of representing the optimal Q function) and the model capacity is much more than needed for this representation, what effect does promoting expressiveness have in terms of learning to solve this problem from finite samples ?
> >
> > For the work to be more convincing I suggest that the authors (1) provide a more principled and analytical motivation for promoting expressiveness (2) provide empirical results on a domain where the optimal state representation is known, ideally with a wide range of task complexity.

---

### Official Review · AnonReviewer3 · 2018-11-02
**Interesting but many elements are lacking or unclear**

**Rating:** 4
**Confidence:** 3

**Review:**

The paper aims at characterizing and discussing the impact of the expressivity of a state representation on the performance. It then discusses three possible ways of improving performance with a specific regularization on the state representation. Overall the idea
of studying the impact of expressiveness and comparing different ways of ensuring it is interesting. However, some parts in the paper are not well-supported and many important elements to understand the experiments are lacking.

Some elements are not well supported and probably not true as stated.
"For fixed h, the expressiveness is related to the MDP M: 1) if the transition p(xt+1|xt, at)π(at|xt) of the MDP is not ergodic, i.e., it can only visit a subset of observations, then the matrix {h(X1 ), · · · , h(Xt ), · · · } will more possibly be low rank; 2) if the reward is very sparse, the representation matrix will be low rank."
If you compare two MDPs, where the first one has two states and is ergodic, while the second one has many more states but is not ergodic, do you think your consideration still applies? For a given h, why would the sparsity of the reward function plays any role?

The paper states that the "method can be widely applied to current DRL algorithms, with little extra computational cost." However, it is not explained clearly how the regularization is enforced. If a supplementary loss is used and minimized at each step, the computational cost is not negligeable.

In the experiment section, the protocol is not clear. For instance the coefficient \alpha is apparently set differently for some of the games? How are they chosen? And why are the hyper-parameters chosen not given?
"However, as the observation type and the intrinsic reward mechanism vary considerably for each game, we also use some other regularization term and coefficient for some of games."

Why are there no details related to the NN architecture?
In Table 1, what does it mean "Times Better" and "Times Worse"?


Other comments:
- In Definition 1, what does it mean X_t \sim \mathcal M? \mathcal M is an MDP, not a distribution. Are the X_t taken following a given policy? are they taken on a given trajectory sequentially or i.i.d.?
- (minor) "From the above description, it is easy to see that the performance of an RL agent depends on two parts. First, it depends on whether the state extractor is good. With a good state extractor, the representation which is a depiction of the observation will retain necessary information for taking actions. Second, it depends on the accuracy of the policy: whether the feed-forward model can correctly take the optimal action given the state." Do the two elements in that paragraph mean the same: "A good state extractor provides an abstract representation from which a performant policy can be obtained?"
- Figure 2: The name of the ATARI game is not mentioned.
- In section 2.2, the MDP framework is introduced (with a state space \mathcal S) but there is no mention of the concept of observation x that is used throughout afterwards.
- In the conclusion, it is stated that "Experiments of A3C and DQN on 55 Atari games demonstrate that ExP DRL can promote their performances significantly." However, the algorithm is not tested on 55 games with DQN.
- The related work discusses papers about state representation but even more directly related to this paper, other papers have also discussed the importance of disentangled representation or entropy maximization for deep RL: https://arxiv.org/abs/1707.08475, https://arxiv.org/abs/1809.04506, ... And papers that discuss expressiveness in deep learning such as https://arxiv.org/pdf/1606.05336.pdf should also be discussed.
- There are many typos/english errors.

---

> ### Author Response · Authors · 2018-11-25
> **To Reviewer2**
>
> Thanks for your very careful reviews!
> Q1: Some comments about the definition of Expressiveness.
> 	a) "Some elements are not well supported and probably not true as stated. ..."
> 	b) "In Definition 1, what does it mean X_t \sim \mathcal M? ..."
> 	c) "... there is no mention of the concept of observation x that is used throughout afterwards."
> A1: For the definition of the expressiveness, we have refined it in the updated pdf in the following aspects: 1) we assume that X_t are taken on a given trajectory; 2) when we say the ergodicity of the MDP matters the expressiveness, we compare two MDPs with the same size of observation space; 3) the discussion about sparse reward is a building issue and has been deleted; 4) we replaced the “state space” to “observation space”.
>
> Q2: "It is not explained clearly how the regularization is enforced. ..."
> A2: Yes, the regularization is used every time the model is updated. The extra computational cost of our proposed algorithm is brought by the SDV decomposition, which is not a dominant term in RL algorithm compared with back-propagation process. We select the game VideoPinBall (the one with highest reward), and find out that the running time for baseline and our proposed method are mostly the same. Based on this, we say “little extra computational cost.”
>
> Q3: "In the experiment section, the protocol is not clear. ..."
> A3: The hyper parameter are chosen as following: 1) we first choose the best alpha for two games from 1 to 10^{-6}; 2) then we apply the best alpha to all games and found that it performs well in most of the games; 3) next we fine tune the games which are not perform well. We have added the hyper parameter settings for every game and an ablation study on \alpha in Appendix B in the new version.
>
> Q4: Details related to the NN architecture
> A4: The NN architecture we used is the same as the one used in paper [1][2]. We have added the description of NN architecture in Appendix A.
>
> Q5: "... Do the two elements in that paragraph mean the same: ..."
> A5: No, here we want to emphasis that the performance of an RL agent depends on two parts, which are the state extractor and the policy learning part. Then we focus on the first part in this paper.
>
> Q6: Related works
> A6: Thanks for pointing out. We added some discussion about these works in Section 5 in the new version.
>
> Q7: Some comments about descriptions in the paper.
> 	a) "Times Better" and "Times Worse"
> 	b) Figure 2: The name of the ATARI game is not mentioned.
> 	c) "In the conclusion, ..."
> A7: We have revised these problems in our updated version.
>
> Q8: There are many typos/english errors.
> A8: We will try to fix them as many as possible.
>
> [1] Peysakhovich, Alexander, and Adam Lerer. "Consequentialist conditional cooperation in social dilemmas with imperfect information." arXiv preprint arXiv:1710.06975 (2017).
> [2] Isakadze, Zurabi. "Towards More Human Like Reinforcement Learning."

---

> > ### Comment · AnonReviewer3 · 2018-12-09
> > **Improved version but two main problems remain**
> >
> > The work has been improved on a few different parts. However, I believe that at least two main problems remain:
> >
> > 1. Concerning the hyper-parameters, the need for choosing a specific regularization form and a specific alpha for every game is not a good sign about the robustness of the results obtained. Ideally, hyper-parameters should be the same for all the games if one want to confidently assess that the improved results do not solely come from added hyper-parameters and "overfitting" of the method to each particular game. If that is not possible, a discussion about why some games require a higher alpha or a specific form of the regularization could be insightful.
> >
> > 2. The related work section has a few additional citations but it has not clarified how previous works have already discussed the importance of a disentangled representation nor how this work positions itself related to these works.
> >
> > Minor: there are still many typos, for instance:
> > - header Table 6 "baeline"
> > - section 4.4 "socres"
> > - ...
> >
> > Minor: It's a bit strange not to have the same number of games for DQN and A3C (30 and 55).
> >
> > Overall, I believe that this work is interesting but it would benefit from a more cautious analysis.

---

### Official Review · AnonReviewer1 · 2018-11-03

**Rating:** 6
**Confidence:** 4

**Review:**

# Summary
This paper proposes a simple regularizer for RL which encourages the state representations learned by neural networks to be more discriminative across different observations. The main idea is to (implicitly) measure the rank of the matrix which is constructed from a sequence of observations and state feature vectors and encourage the rank to be high. This paper introduces three different objectives to implement the same idea (increasing the rank of the matrix). The experimental results on Atari games show that this regularizer improves A3C on most of the games and show that the learned representations with the proposed regularizer has a high rank compared to the baseline.

[Pros]
- Makes an interesting point about the correlation between RL performance and state representations.
- Proposes a simple objective function that gives strong empirical results.

[Cons]
- Needs more empirical evidences to better support the main hypothesis of the paper.

# Novelty and Significance
- This paper makes an interesting observation about the correlation between the RL performance and the how discriminative the learned features.
- To verify it, this paper proposes a new and simple regularizer which improves the performance across many Atari games.

# Quality and Experiment
- The main hypothesis of this paper is that the expressiveness of the features (specifically the rank of the matrix that consists of a sequence of features) and its RL performance is highly correlated. Although this paper showed some plots (Figure 2, 6) to verify this, a more extensive statistical test or experiments would be more convincing to show the hypothesis. Examples would be:
1) Measuring the correlation between the two across all Atari games.
2) An ablation study on the hyperparameter (alpha).
3) Learning state representations just from the reconstruction task (without RL) with/without the proposed regularizer and separately learning policies on top of that (with fixed representations). It would be much more convincing if the regularizer helps even in this setup, because this would show the general effect of the expressiveness term (by removing the effect of RL algorithm on the representation).
- This paper only reports "relative" performances to the baseline. Though it looks strong, it is also important to report the absolute performances in Atari games (e.g., median human-normalized score, etc) to show how significant the performance gap is.
- The results with DQN are not convincing because the agents are trained only for 20M frames (compared to 200M frames in many other papers). It is not much meaningful to compare performances on such a short training regime. I would suggest running longer or removing this result from the paper and focusing on more analysis.

# Clarity and Presentation
- Figure 1a is not much insightful. It is not surprising that the representations that led to a poor policy (which achieves 0 reward) are much less discriminative given five situations with five distinct optimal actions, because the the policy has no idea which action is better than the other in such situations. It would be more informative to pick N consecutive frames and show how scattered they are in the embedding space.

---

> ### Author Response · Authors · 2018-11-25
> **To Reviewer1**
>
> Thank you for the helpful comments. Here are our responses.
> Q1: A more extensive statistical test or experiments to convince the hypothesis:
>     a) Measuring the correlation between the two across all Atari games.
>     b) An ablation study on the hyperparameter (alpha).
>     c) Learning state representations separately…
> A1: a) Curves tracking testing rewards and the expressiveness in Fig. 2 and Fig. 6 have demonstrated their correlation over multiple Atari games, which have shown some statistical significance. In order to make the results more convincing, we have added the study about relationship between model performance and representations across all Atari games in Section 2.2.1 in our updated version.
> b) The ablation study on hyperparameter (alpha) is shown in Appendix B now, which shows that improvements are stable with different alpha.
> c) Our proposed method is designed and applied to RL algorithms. Although one RL agent model can be viewed as the state extractor and the policy learning part, they are trained end-to-end within the RL algorithm loop. These two parts influence each other during training. Besides, we do not emphasis which part works in our proposed method indeed, we just make a correlation between performances and the expressiveness of representations.
>
> Q2: Detailed questions for the experimental results.
> A2: 1) We also provide the absolute performances of all Atari games in Appendix C in the updated version. 2) For DQN, we finished the experiments for 200M frames and updated the results in Section 4.4 in the new version.
>
> Q3: "Figure 1a is not much insightful. ... "
> A3: Fig. 1a has been refined in the following way: selected frames and their corresponding embedding points are removed.

---

> > ### Comment · AnonReviewer1 · 2018-12-10
> > **Thanks**
> >
> > Thanks for addressing my questions. The revised paper seems to provide a bit more evidence about the relationship between expressiveness and performance in RL. One thing that I realized is that the authors used different hyperparameters (rank regularization term, alpha) for each Atari game, which makes the proposed algorithm look less practical. I decided to keep my score (6).
> >
> > One minor comment: Please compute "human"-normalized score in Table 5 and 6.

---

### Author Response · Authors · 2018-11-25
**Summary of the new version**

Thanks for all reviewers and their comments, which are helpful for us. We have uploaded a new version of our paper, and the main changes include:
+ The observation section 2.1.1 is renewed, and observations across all Atari games are added.
+ The model architecture is shown in Appendix A
+ All hyper parameters used are listed in Appendix B
+ An ablation study on alpha is shown in Appendix B
+ All raw scores and normalized scores are listed in Appendix C
+ Experiments of L2 norm regularizer is added in Appendix D

---

### Meta-Review · Area_Chair1 · 2018-12-14

**Confidence:** 4
**Recommendation:** Reject

**Metareview:**

The authors propose to define 'Expressiveness' in deep RL by the rank of a matrix comprising a number of feature vectors from propagating observations through the learnt representation, and show a correlation between higher rank and higher performance. They try 3 regularizers to increase rank and show that they improve the final score on Atari games compared to A3C or DQN. The AC and reviewers agree that the paper is interesting and novel and could have general significance for the RL field. Also, the authors were very responsive to the reviewers and added more details, plus several experiments and analyses to support their claims. However, the reviewers were concerned about a number of aspects and have recommended that the authors clean up their presentation and analysis a bit more. In particular, the fact that the regularization coefficient is tuned for each Atari game makes it very hard to compare to DQN/A3C which are very careful to keep the same hyperparameters across every game.